# Measuring Work-Related Functioning Using the Work Rehabilitation Questionnaire (WORQ)

**DOI:** 10.3390/ijerph16152795

**Published:** 2019-08-05

**Authors:** Monika E. Finger, Reuben Escorpizo, Alan Tennant

**Affiliations:** 1Participation, Integration & Social Epidemiology Group, Swiss Paraplegic Research, 6207 Nottwil, Switzerland; 2ICF Research Branch, WHO FIC CC Germany (DIMDI) at SPF, 6207 Nottwil, Switzerland; 3Department of Rehabilitation and Movement Science, University of Vermont, Burlington, VT 05452, USA

**Keywords:** vocational rehabilitation, work, WORQ, International Classification of Functioning Disabilityand Health (ICF), functioning, Rasch analysis

## Abstract

The assessment of work-related functioning is a key process in vocational rehabilitation to identify specific domains of disability that can be considered within return to work strategies. The Work Rehabilitation Questionnaire (WORQ) was developed to evaluate work-related functioning based on the International Classification of Functioning, Disability, and Health (ICF) framework and is available in different languages. The aim of this study was to assess the French version of the WORQ using item response theory to further validate the scale. Rasch analysis of WORQ and the WORQ-BRIEF (a brief version of the WORQ) was performed using a calibration sample of 221 persons with musculoskeletal injuries. A four-testlet solution indicated the unidimensionality of WORQ, with no differential item functioning for age, education, physical job demands, and injury severity. Reliability was 0.969 and 0.918 for WORQ and WORQ-BRIEF, respectively. The minimal detectable change was calculated to be 4.2% of its operational range for WORQ and 8.5% for WORQ-BRIEF. Consequently, the French version of WORQ can be considered a good measure of work-related functioning in musculoskeletal conditions. WORQ can be used in rehabilitation practice to comprehensively identify the disability and guide clinical decision making and intervention planning. Further studies are needed to evaluate the psychometric properties of WORQ in other health conditions.

## 1. Introduction

Work participation is considered as a major indicator of social participation for persons in working age [1,2]. In general, and particularly for those with disabilities, work is associated with economic self-sufficiency and enhances psychological well-being and self-worth, personal identity, and quality of life [3]. Work participation also assures social integration [4]. From a societal perspective, particularly with an aging workforce and, at the same time, a demanding and rapidly changing work environment that requires high work performance and flexibility, assuring employment for persons with disabilities becomes increasingly important but also challenging [5]. Associated increases in sickness absence, a decline in work productivity, along with an increase in long-term disability allowances further adds to social security costs [6,7]. As a countermeasure, in the last decades, national and private insurances along with governmental social security systems increased their efforts to provide appropriate management of disability to support timely return to work and prevent workers with disability from early retirement [8,9,10].

A careful assessment of the workers’ health problems and available resources is critical in disability management, of which vocational rehabilitation is often a key part. Research confirms a favorable return-to-work outcome by actively involving the patient in the process, e.g., evaluating his or her expectations, formulating goals, and selecting interventions [11,12,13]. To strengthen person-centeredness in this process and to adhere to good practice guidelines and policy recommendations, clinicians increasingly use patient-reported outcome measures (PROM) “to measure what matters to the patient” [14,15]. In addition, PROMs allow us to identify and quantify multi-faceted problems in functioning—such as those that affect body functions and structures, activities, and participation given the context of work tasks and the work environment—in a more cost-effective manner compared to clinical testing of work functioning [16,17,18,19].

In this context, professionals use a growing number of PROMs that were often developed for specific health conditions or for a distinct work population or environment [20,21,22,23]. Nevertheless, a psychometrically sound generic functioning questionnaire that is applicable across health conditions or vocational rehabilitation settings would be crucial to be able to compare the impact of problems in functioning across diseases and along the disability management process. One such measure is the Work Rehabilitation Questionnaire (WORQ). The self-reported WORQ is based on the International Classification of Functioning, Disability, and Health (ICF). WORQ was designed to assess and document the most relevant domains related to body function, activities, and participation in the context of work-related functioning and vocational rehabilitation [24,25]. Those relevant domains were chosen from the ICF Core Set for vocational rehabilitation, which refers to a list of ICF categories (i.e., aspects of functioning) selected through an evidence-based process to describe the most relevant functioning aspects of persons undergoing vocational rehabilitation [26]. WORQ was demonstrated to measure change in work-related functioning over time based on a patient’s experience. The full version of WORQ consists of 40 items on work-related functioning and contains four clinical subscales addressing emotion, cognition, dexterity, and mobility and 10 additional single items. WORQ captures the extent of functioning problems for all 40 items with a numeric rating scale of 0 to 10, on the basis of which the calculation of an average total score was proposed [27,28]. Based on the full version of WORQ, a BRIEF version (WORQ-BRIEF) was developed, which contains 13 items; 10 items represent the four clinical subscales, and three additional functioning items—energy and drive, pain, and relationships with people—were statistically identified in earlier studies as most relevant to describe problems in functioning for persons with any health condition [29,30].

There are multiple language versions of WORQ, all of which have undergone a formal cross-cultural adaptation process [25]. Using classical psychometric methods, the French language WORQ and WORQ-BRIEF both showed excellent test-retest reliability [intraclass correlation (ICC): WORQ = 0.935; WORQ-BRIEF = 0.913] and scale reliability (Cronbach’s alpha: WORQ = 0.968; WORQ-BRIEF = 0.955). WORQ showed good construct validity with General Functioning (Pearson correlation: 0.66), General Health (Pearson correlation: 0.48), and the Hospital Anxiety and Depression scale (HADS) (Pearson correlation: 0.56).

However, if WORQ is to be used as outcome measure longitudinally, valid change scores are needed, and if WORQ scores are to be compared between persons, different health groups, and settings, a summary score based on an interval metric becomes indispensable [31]. However, although the development of WORQ was conceptually based on the model of the ICF and the ICF core sets for vocational rehabilitation (VR), it remains unclear if all 40 items of WORQ and the 13 items of WORQ-BRIEF reflect a single concept of work-related functioning; this assumption is a prerequisite to a reliable interval score [30]. A further requirement that clinicians can reliably use a generic questionnaire such as WORQ across different health conditions and settings is that patients have to understand and answer the items in a similar way independent of their age, gender, education, or other distinctive attributes (invariance) [32].

Modern psychometric approaches, particularly those associated with the Rasch measurement model, provide additional evidence about the internal construct validity (unidimensionality), the reliability, and the extent of invariance of scales, which also allows for transformation of the raw scores from ordinal data into interval data or scores [33].

Hence, the overall objective of this study was to examine the scale validity of the French version of the WORQ and the WORQ-BRIEF using Rasch analysis. The specific aims of this psychometrics study were (1) to determine if the items of WORQ-French and WORQ-French-BRIEF measure a common overarching concept—work-related functioning—and therefore can be presented in total sum-scores, (2) to determine if the four clinical domains of WORQ (cognition, emotion, dexterity, and mobility) are reliable and valid to offer greater granularity of interpretation and, (3) to provide interval transformation tables for the total-scores and the domains of WORQ and of the WORQ-BRIEF.

## 2. Materials and Methods

### 2.1. Subjects, Setting, and Eligibility

We collected the data from 221 participants in a longitudinal observational study in a single rehabilitation teaching hospital in Switzerland. Participants with musculoskeletal injuries (MSK) undergoing in-patient rehabilitation were recruited from French-speaking Cantons of Switzerland. The study (CCVEM 005/15) was approved by the ethical committee of the canton Valais (ICHV) and conducted according to the principles outlined in the Declaration of Helsinki.

The full version of WORQ-French was collected at five time points from February 2015 to December 2016: admission to the center of VR (T0), seven days after admission (T1), at discharge (T2), three months post-discharge (T3), and six months post-discharge (T4).

Participants were recruited consecutively using convenience sampling, i.e., all patients were checked for their inclusion criteria when they entered the department of vocational rehabilitation. To be eligible for this study, the participants (1) had to be receiving at least one vocational rehabilitation intervention irrespective of their type of MSK, (2) were able to speak and read French fluently, (3) were between 18 and 65 years old, (4) were employed or looking for a new job, and (5) signed an informed consent prior to participate in the study.

### 2.2. Outcome Measure: WORQ French Version and WORQ-French-BRIEF

The full version of WORQ and the BRIEF version have two parts, where part one is identical for both versions. This part contains 17 sociodemographic and work-related questions. The work-related questions include profession, work status, work demands, VR interventions, and the amount of support received by family, employer, and labor and employment services. Part two of the full version consists of 40 items on functioning (18 body functions and 22 activities and participation items), whereas part two of the BRIEF version consists of 13 items (6 body functions and 7 activities and participation items). WORQ is freely available from www.myworq.org [25]. A sum-score of the items was calculated.

In addition, four single domains (a domain is any meaningful aggregation of ICF categories) were identified in the full version of WORQ based on a previous factorial analysis [34,35]. These domains are: Emotion, Cognition, Dexterity, and Mobility (Table 1). These domains are intended to identify a specific underlying pattern of functioning, which we expect will aid clinical decision making and intervention allocation in VR. Ten items are not assigned to any domain; four items are assessing sensory functions and pain, and two items relate to energy and sleep, where the remaining four items relate to skin problems, transport, relationship, and covering costs of living. These items were considered relevant in the development process of WORQ to complement the picture of work-related functioning and to take into account the different needs of the patients.

### 2.3. Hypothesis

We hypothesized that WORQ full version and WORQ-BRIEF assess a unidimensional concept of “work-related functioning”, and therefore a total sum-score could be calculated for both versions accordingly. We also hypothesized that the four domains would also be confirmed in a similar manner.

### 2.4. Statistical Testing

Descriptive statistics was used to describe the sample characteristics including age, gender, family status, education, injury location, and physical job demands (rated qualitatively by a health professional as high, moderate, or low physical job demand). Descriptive analyses were performed with the software package IBM SPSS Statistics for Windows, Version 24.0 (Armonk, NY, USA) [36].

The Rasch model was used to test the scale validity and to create interval measurement scales for WORQ, WORQ-BRIEF, and the domains [37]. Using the partial-credit Rasch model [37] with the RUMM 2030 software (RUMM Laboratory: Perth, WA, Australia) [38], a good fit of the data to the Rasch model was sought, confirming unidimensionality and making it possible to transform ordinal scale observations into interval scale measures.

#### 2.4.1. Dataset

Initially, we constructed a dataset called “calibration sample” that contained the 40 questions from part two of the full version of WORQ. No person was entered into the calibration sample with more than one time point (T0–T4). This approach corrects for potential time effects, and applying a “calibration sample” legitimizes the use of a questionnaire in longitudinal studies [39]. The calibration sample was used to analyze WORQ, WORQ-BRIEF, and the clinical subscores.

#### 2.4.2. The Rasch Model

By applying the data to the Rasch partial credit model, four principal requirements were tested, proof of which would deliver a valid Rasch-based scale with interval-scale properties. These requirements are monotonicity, local item independence, unidimensionality, and invariance of groups, both of those groups with different levels of functioning and also of groups defined by various contextual factors such as gender or health [40,41,42]. These requirements were tested through fitting data to the model, a process referred to as Rasch analysis and described in detail elsewhere [28]. Ideal values for fit are given at the bottom of the fit table. If all requirements of the Rasch model were satisfied, it produced an interval scale latent estimate upon which both item difficulties and person difficulties (in this case, their level of functioning) were placed. In addition, the reliability of WORQ was examined by the person separation index (PSI), and the targeting of the scale (e.g., floor and ceiling effects) was examined.

Certain aspects of Rash analysis were recently updated and adjust the methodology given above. It was shown that the test for the (conditional) independence of items using the standardized residuals should set the threshold for a breach of that independence at 0.2 above the average residual correlation. Where local item dependencies are observed, they can be aggregated to enter the analysis as testlets (if based upon existing domains) or “super-items” otherwise [43,44,45]. When testlets are used, the RUMM 2030 software enacts a bi-factor equivalent solution and reports the proportion of common variance retained in the data in order to provide a unidimensional latent estimate [46]. This proportion should be 0.9 and above if the scale is to be considered unidimensional [47].

Given adequate fit to the model, transformation tables convert the ordinal sumscores to interval scores. To transfer the raw scores to an interval-level scale of the same range as the original raw scores, the metric (in logits) derived from the Rasch analysis was linearly transformed.

#### 2.4.3. Standard Error of Measurement

Based on the transformed Rasch scores, the standard error of measurement (SEM) was calculated, representing the amount of error that indicates the amount of variability in a test administered to a group caused by measurement error. We calculated SEM using the Rasch reliability estimate (PSI) as the reliability coefficient Rx; SEM = SD $\sqrt{1-\mathrm{Rx}}\;$ [48]. The minimal detectable change (MDC), meaning the minimum amount of change in a patient’s average score that is not the result of measurement error, was calculated on the 95% probability as MDC95 = 1.96 × SEM × $\sqrt{2}\;$ [49].

## 3. Results

Two hundred and twenty-one patients participated in the study. The majority of participants (89%) were male with a mean age of 43 years (SD 10.9); 52% of the participants had finished high school, 24% primary school or less, and 24% reported to have a college or university degree including post graduate education. Of the patients, 30% reported light physical work, 47% reported moderate, and 23% reported severe physical work. The majority of participants suffered from an injury to the extremities (39% upper extremity, 37% lower extremity). Only 11% of the participants suffered from a back problem and 13% from poly-trauma (Table 2). The final calibration sample contained data from all five administration time points: 50 datasets from T0, 26 from T1, 49 from T2, 46 from T3, and 50 from T4.

A first Rasch analysis was performed for the full version of WORQ with all 40 items. Substantial local dependency (LID) was found with 107 pairs of items with residual item correlations >0.2. The LID led to a considerable deviation from the model, and multidimensionality was detected. Threshold disordering was also evident, predominantly in items with LID. A subset of items was chosen and fit to the model, where disordered thresholds mostly disappeared irrespective of LID in the full item set. Therefore, it was assumed that LID was causing the distortion of threshold ordering and that recoding the categories would only confuse the users. No further action was taken other than to proceed with dealing with the LID.

In a first step, two testlets were created to account for two pairs of interdependent items. The first pair was Q21 “… lifting and carrying objects weighting up to 5 kg” and Q22 “… lifting and carrying objects weighing more than 5 kg”, and the second pair was Q30 “… walking a short distance” and Q31 “… walking a long distance.” In a next step, four testlets were created, including the two testlets from step one and all remaining 36 items. Thus, all four testlets scored 0–100. Then, the data were reanalyzed. The four-testlet solution satisfied the Rasch model with strict unidimensionality (3.8% of significant t-tests) (Table 2). No further LID was present, and no differential item function (DIF) by age, level of education, physical job demands, injury severity, or time point of data collection was detected. DIF was not analyzed for sex because of the disproportionate data (only 11% of the participants were female).

Unidimensionality and fit to the Rasch model based on testlet solutions were also validated for the total sum-score of WORQ-BRIEF and for all four clinical subscores (Table 3). DIF for age, physical job demands, and education remained present only in the Mobility subscore. This finding may have been due to having only four items in the score. With a PSI above 0.9 for the full version of WORQ as well as WORQ-BRIEF and a PSI above 0.85 for the four clinical subscales, all scales could be used for individual measurement. The fact that the A-value stayed close to one showed that all the variance in the data was retained. Detailed information on the testlet solutions can be found in the Appendix A.

The targeting of items to the population could be considered as good to very good, with a person location of −0.506 (SD 0.613) logits for the WORQ scale and −0.286 logits (SD 0.643) for the WORQ-BRIEF scale, where both scales had an average location of zero logits. Hence, the average for the person location lying below zero indicated that the participants in this study had slightly fewer problems than in a perfectly targeted population. The slight deviation could be expected because WORQ, designed as a generic questionnaire, contains items that are relevant to cover the overall spectrum of work-related functioning in diverse populations but are not necessarily specific to a single health condition. See Figure 1.

Moreover, no ceiling or floor effect was detected in any of the scales (Table 4).

As all tested scores fulfilled the requirements of the Rasch model, transformation tables with the raw scores and the corresponding interval scores were created. They can be accessed in the Appendix A.

The minimal detectable change (MDC) was calculated based on the interval scale-metric (Appendix A). With an MDC95 of 4.23% or 0.42 points on the 0–10 scale, the error attributed to WORQ was minimal compared to other patient reported questionnaires [50]. The small MDC might have been attributed to 40 items of which a substantial number scored zero in our population. WORQ-BRIEF, on the other hand, had 8.47% or 8.5 points on the 0–10 scale, a comparable MDC to most other questionnaires. The subscores, with MDCs between 13% and 20%, indicated that a substantial change in the scale is needed to take into account a real difference.

## 4. Discussion

This study examined the scale reliability and the validity of WORQ, WORQ-BRIEF, and four WORQ subscores regarding fit to the Rasch model. The findings support that WORQ is assessing an overarching concept, which is work-related functioning. All tested scales proved very good to acceptable measurement properties in terms of DIF and unidimensionality. A reliability (PSI) of more than 0.86 for all the scales confirms that we can strongly relate on the fit characteristic and confirms the use of WORQ not only for clinical decision making but also for measurement of change in work-related functioning [51].

From the beginning, WORQ aimed to elaborate the complex problems in work-related functioning of persons in VR, which led to selecting WORQ items from the ICF core set for VR based on statistics, literature search, and clinical expertise. When applying the Rasch model with a testlet solution, after LID was removed, the data fully satisfied model requirements, including invariance of contextual factors [52]. As such, most of the variance in WORQ was still retained, strongly supporting the analysis approach, which crucially avoided the need to first rescore the 0–10 scales of items with disordered thresholds that were most likely artificial, being caused by LID [53]. Retaining the original 0–10 numeric rating scale scoring for all items also enhances the intuitive understandability of WORQ in clinical practice and rehabilitation [54].

Further, the additional information gained from the four clinical subscores may allow clinicians and researchers to better understand underlying functioning patterns and help to identify subgroups with specific areas of needs within a sample [55].

The consistent response patterns across different levels of injury severity, education, physical job demands, or age for all WORQ scores demonstrate that WORQ is not prone to DIF for the tested factors in our mixed musculoskeletal (MSK) group. The finding supports measuring and comparing similar groups of patients based on work-related functioning measured with WORQ. Nevertheless, evaluating the invariance of WORQ in persons with other than primarily physical impairments, for example, those with mental health conditions, will need to be undertaken in relevant populations.

We confirmed the scale reliability of WORQ and WORQ-BRIEF, which indicates that clinicians can utilize the WORQ items [ordinal (raw) values] in a total sum-score to monitor progress in a single person; however, when calculating change scores, only the direction of change is obtained, not its magnitude. However, transforming ordinal values to interval-based latent estimates (as shown in the Appendix A) is a natural and fast procedure. The direct transformation of the original (WORQ raw score: 0–400) scale to an interval scale of the same range (WORQ interval: 0–400) enables a direct comparison of the effect of the transformation. Our results thereby confirm the findings of Forrest and Andersen, who noted as early as 1986 [56] that the values in cumulative patient reported questionnaires are tied to an artificial range (e.g., WORQ summary score 0–400) that causes distortion of intervals towards the margins. For example, raw numbers of 15–45 points on the ordinal sum-scores scale from WORQ (0–400 scale) result in a 30 point change. These two raw scores are transformed to 101 and 128 points on the interval scale and result in a similar change of 27 points. On the other hand, focusing on the same 30 point range near the center of the ordinal scale, say, scores of 200 and 230 points, translates to interval scores of 183 and 191; an interval change score of only eight points. This finding indicates a much smaller change on the interval scale compared to the ordinal raw score. Interval scales are characterized by measurement units that maintain the same size across the entire scale, resulting in a more accurate measurement of change than an ordinal scale. Transforming ordinal scores to interval scores may not only be relevant when calculating change scores but may be even more so when comparing changes in functioning between individuals or groups of individuals in different settings. A further benefit of interval transformed scores is that researchers who want to include data on work-related functioning into their analyses are now able to use parametrical statistics (given appropriate distributions). However, since the “specific objectivity” of the Rasch model indicates that the results are specific to the sample, the transformation tables in this article refer specifically to MSK problems [57]. Consequently, it is an empirical mater as to whether or not the existing transformation holds in other conditions (thus making a true generic scale).

### 4.1. Limitations

This study has some weaknesses. Geographical and cultural nuances of the study sample may limit the transferability of the results to other MSK populations, although individuals with multiple cultural backgrounds and various injury types, time points since injury, or professions were included. Nevertheless, the more significant concern may be that less than 10% of the population were female, as gender is known to lead to DIF in a substantial part of self-reported questionnaires [58]. Hence, clinicians and researchers should consider biological or gender differences when applying WORQ in a mixed or predominantly female population. The sample was recruited with convenience sampling, which also limits the representativeness of the results to the population.

Although the final testlet solution for WORQ satisfied all Rasch model requirements, the four-testlet strategy for WORQ was somewhat forced by the restrictions of the data analysis software RUMM 2030 that only allows testlets with a maximum number of 100 thresholds, e.g., ten items rated with a 0–10 rating scale. This software restriction also prevented us from testing a two testlet solution for the full version of WORQ and for a testlet solution including the clinical subscores (which would have provided more robust fit statistics) [59]. Luckily, this restriction proved unimportant for the current analysis, as the final solutions showed excellent model fit and no marginal drop in PSI for WORQ or WORQ-BRIEF, hence no further testing was required.

### 4.2. Clinical Use of WORQ, WORQ-BRIEF, and Clinical Subscores

While WORQ was initially designed as a questionnaire to detect and understand problems in work-related functioning from the client’s perspective, the findings of this study suggest that WORQ may also be used to measure reliable change in work-related functioning of individuals and groups in MSK VR [60]. The subscores may be used to better explore the extent of one of the underlying traits: cognition, emotion, dexterity, or mobility. They may also help to disentangle the complex construct of work-related functioning in the decision-making process, help to closer monitor change in specific areas of functioning, and therefore allow one to address the needs of these groups more specifically (Figure 1b).

WORQ-BRIEF, on the other hand, can be used as a fast screening questionnaire of the major areas of WORQ related problems. It is debatable if WORQ-BRIEF is more suitable in research than WORQ. On the one hand, it is much shorter, but WORQ-BRIEF is also less sensitive according to the MDC and the reliability measured by the PSI.

Although WORQ was initially designed for the use in vocational rehabilitation, newer findings show that WORQ is also reliable in detecting the needs of patients with small levels of work-related functioning in physiotherapy outpatient settings [61]. Based on this finding, it should be further evaluated whether WORQ could also be used in public health in the context of health prevention and to support sustainable employment. Sustainable employment characterizes a person–job–workplace match that enables a person to stay healthy and satisfied at work over time, with a work performance that meets the expectations of the person and the employer [62]. A focus on prevention of work related disability is becoming increasingly important, as an increasing number of workers experience a decline in work-related functioning while aging. An instrument that detects change in functioning early may help to identify those who need support and to prevent reduced work ability or early exit from labor market. Using a generic instrument such as WORQ to monitor the level of functioning in workers at risk may also follow the notion of the Organisation for Economic Co-operation and Development (OECD) that patient reported outcome measures are key to facilitating high-value care to gain a complete picture of what happens to people across the pathway of care [63].

## 5. Conclusions

In conclusion, the French version of WORQ, WORQ-BRIEF, and the clinical subscores are reliable and well targeted measures of work-related functioning. WORQ can be used in clinical practice to reliably measure change and comprehensively identify problems in work-related functioning in MSK. The clinical subscores may help to guide clinical decision making and intervention planning in VR or occupational health. Transformation tables (Appendix A) enable the calculation of change scores to document reliable change in functioning, while clinical subscales may help to detect underlying patterns in functioning. With only 13 items, WORQ-BRIEF may be more practical to be used in research or to compare populations across health conditions compared to WORQ. Further studies are needed to evaluate psychometric properties of WORQ in other health conditions with a specific focus in populations with mental problems or multiple comorbidities and in prevention activities.

## Figures and Tables

**Figure 1 ijerph-16-02795-f001:**
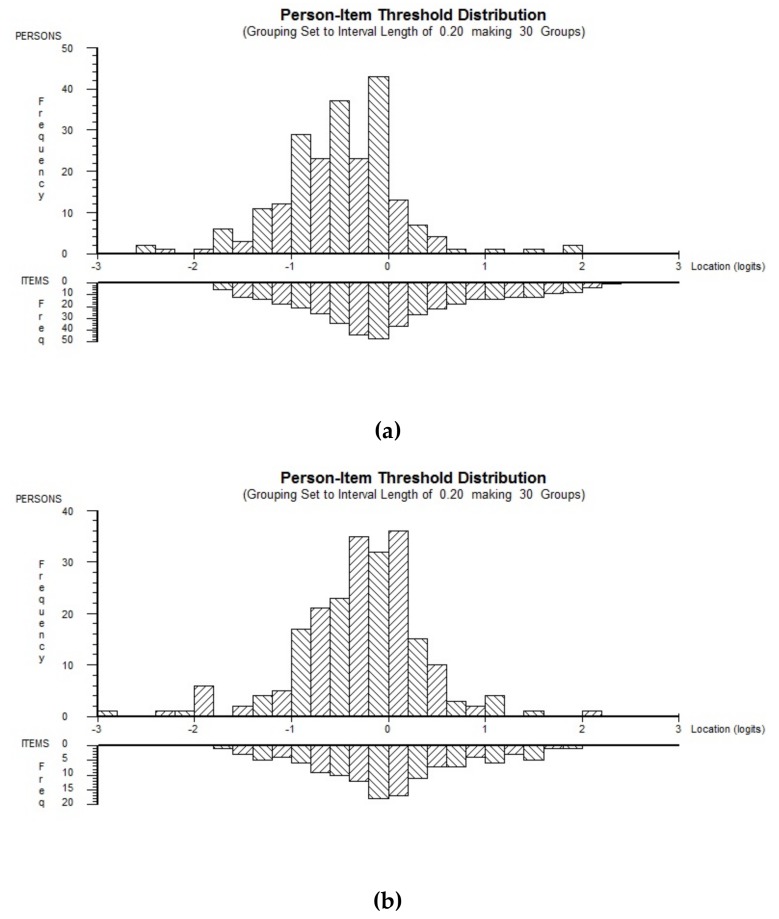
Person-Item Threshold Distribution: (**a**) WORQ and (**b**) WORQ-BRIEF.

**Table 1 ijerph-16-02795-t001:** Items of The Work Rehabilitation Questionnaire (WORQ) subscores.

Overall in the Past Week, to What Extent did You Have Problems with…		ICF	E	C	D	M
3	… remembering to do important things?	BF		✓		
4	… your usual daily activities because you felt sad or depressed?	BF	✓			
5	… your usual daily activities because you felt worried or anxious?	BF	✓			
6	… being irritable?	BF	✓			
7	… your temper?	BF	✓			
8	… your self-confidence?	BF	✓			
9	… thinking clearly?	BF		✓		
10	… analyzing and finding solutions to problems in day to day life?	BF		✓		
12	… keeping your balance while maintaining a position or during movement?	BF				✓
14	… general endurance when performing physical activities?	BF			✓	
15	… muscle strength?	BF			✓	
17	… learning a new task (e.g., learning a new game, learning how to use the computer, learning how to use a tool, etc.)?	A&P		✓		
18	… focusing attention on a specific task or e.g., filtering out distractions such as noise?	A&P		✓		
19	… reading?	A&P		✓		
20	… making decisions?	A&P		✓		
21	… starting and completing a single task such as making your bed or cleaning up your desk or workplace?	A&P			✓	
22	… carrying out your daily routine or day to day activities?	A&P			✓	
23	… handling stress, crises, or conflict?	A&P	✓			
24	… understanding body gestures, symbols and drawings?	A&P		✓		
25	… starting and maintaining a conversation?	A&P		✓		
26	… using communication devices such as using a telephone, telecommunication devices, and computers?	A&P		✓	✓	
27	… lifting and carrying objects weighing up to 5 kg?	A&P			✓	
28	… lifting and carrying objects weighing more than 5kg?	A&P			✓	
30	… walking a short distance (less than 1 km)?	A&P				✓
31	… walking a long distance (more than 1 km)?	A&P				✓
32	… moving around including crawling, climbing, and running?	A&P				✓
34	… driving a car or any form of transportation?	A&P			✓	
35	… getting dressed?	A&P			✓	
36	… looking after your health such as maintaining a balanced diet, getting enough physical activity and seeing your doctor as needed?	A&P			✓	

(ICF = International Classification of Functioning, Health, and Disability, BF = Body functions, A & P = Activities and Participation, E = Emotional subscore, C = Cognitive subscore, D = Dexterity subscore, M = Mobility subscore).

**Table 2 ijerph-16-02795-t002:** Characteristics of study participants.

Characteristics	*n* = 221 (%)
Age, mean, years	43.47 (10.9 SD)
Sex, male	196 (89.0%)
Family status	
single	48 (21.7%)
married/cohabitant	125 (56.5%)
separated/divorced	48 (21.7%)
Injury location	
upper extremity	85 (38.6%)
lower extremity	82 (37.1%)
trunk/back	25 (11.4%)
poly-trauma	29 (13.2%)
Injury score	
minor	58 (26.5%)
moderate	120 (54.8%)
severe	41 (18.7%)
Job severity	
light physical work	70 (30.3%)
moderate physical work	108 (46.8%)
severe physical work	53 (22.9%)
Education	
less than primary school	11 (5%)
primary school	41 (19%)
secondary school	115 (52%)
college or university	47 (21%)
post graduate education	7 (3%)

**Table 3 ijerph-16-02795-t003:** Rasch analysis summary for all tested sum-scores and subscores of WORQ.

Subscale	Number of Items	Stage	Item Fit Residual		Person Fit Residual		χ2			Reliability		LID	Differential Item Function		Unidimensionality: Paired t-Tests			
			Mean	SD	Mean	SD	Inter-action Value	df	*p*	PSI	A		Uni-form DIF	Non-Uni-form DIF	*N* Significant Tests	*N* Sample	Proportion of Significant Tests	%LB 95CI
Work rehabilitation questionnaire (WORQ)	40 items	Start	−0.49	0.66	−0.10	1.91	288.8	80	0	0.97		Yes	No	No	73	220	0.38	0.35
	4 testlets	Final	−0.04	0.72	−0.43	1.05	2	8	0.98	0.97	1.01	No	No	No	8	214	0.04	0.01
Work rehabilitation questionnaire –brief version (WORQ-BRIEF)	13 items	Start	0.25	2.60	-0.13	1.31	138.4	24	0	0.92		Yes	a, i, p, e	No	17	219	0.08	0.05
	3 testlets	Final	−0.16	0.41	−0.48	1.06	8.1	6	0.23	0.92	1.00	No	No	No	6	215	0.03	0.00
Emotion	6 items	Start	−0.04	0.79	−0.65	1.47	10.5	12	0.57	0.90		Yes	a, i, p, e	No	12	208	0.07	0.04
	2 testlets	Final	0.07	0.24	−0.52	0.84	3.08	4	0.54	0.91	1.06	No	No	No	6	207	0.03	−0.00
Cognition	10 items	Start	0.10	1.92	−0.39	1.33	35.4	20	0.02	0.86		Yes	Yes	No	10	192	0.05	0.02
	2 testlets	Final	0.13	0.30	−0.45	0.77	3.6	4	0.46	0.86	1.00	No	No	a **	7	189	0.04	0.01
Dexterity	10 items	Start	0.21	2.35	−0.32	1.26	85.9	20	0.00	0.90		Yes	g;	a, i, p, e	8	217	0.04	0.01
	2 testlets	Final	0.14	0.41	−0.42	0.74	2.8	4	0.58	0.88	0.98	No	No	No	8	214	0.04	0.01
Mobility	4 items	Start	−0.36	2.56	−0.38	0.96	51.5	8	0.00	0.81		Yes	a, i, p, e	No	0	218	0.00	−0.03
	2 Testlets	Final	0.13	0.05	−0.39	0.73	6.5	4	0.16	0.87	1.08	No	a, p, e	No	2	204	0.01	−0.02
Ideal values			<0.5	<1.4	<0.5	<1.4			>0.05	>0.85							0.05	0.05

PSI = person separation indexp; LID = local dependency; DIF = differential item function; DIF Bonferroni corrected. DIF: a = age, i = injury score, p = physical job demands, e = level of education * Diff only for testlet 2; ** Diff for 1st age quartile only (youngest workers).

**Table 4 ijerph-16-02795-t004:** Minimal detectable change and ceiling and bottom effect.

WORQ Versions and Clinical Subscores	*n* Items	SEM	MDC 95 Points %		Ceiling Effect %	Floor Effect %
WORQ sumscore	40	1.53	4.23/100	4.23	0.45	9.50
WORQ BRIEF sumscore	13	3.05	8.47/100	8.47	0.45	0.45
Cognition subscore	10	6.41	17.77/100	17.77	12.67	0.45
Emotion subscore	6	2.90	8.05/60	13.41	4.98	0.45
Dexterity subscore	10	4.66	12.91/100	12.91	0.45	1.36
Mobility subscore	4	2.87	7.96/40	19.90	1.36	4.98

MDC = minimal detectable change, SEM = standard error of measurement.

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
