# Peer review of "Measuring Work-Related Functioning Using the Work Rehabilitation Questionnaire (WORQ)"

_ijerph, 2019, doi:10.3390/ijerph16152795_

Round 1

Reviewer 1 Report

Dear authors,

Thank you for your manuscript of which I read with interest. The aim of this study was to validate the French version of the WORQ and WORQ-BRIEF using Rasch analysis. However, the conclusions do not seem to reflect what the study methodology can set out to achieve (e.g. to conclude that ''WORQ can be safely used in clinical practice''). In addition, this validation study will also benefit from assessing the agreement of WORQ and WORQ-BRIEF with another valid instrument.

Author Response

Dear Reviewer,

Thank you for reviewing our manuscript. Please find attached the point to point review for both reviewers.

Reviewer #1:

Dear authors,

Thank you for your manuscript of which I read with interest. The aim of this study was to validate the French version of the WORQ and WORQ-BRIEF using Rasch analysis.

Comment 1:

However, the conclusions do not seem to reflect what the study methodology can set out to achieve (e.g. to conclude that ''WORQ can be safely used in clinical practice''). In addition, this validation study will also benefit from assessing the agreement of WORQ and WORQ-BRIEF with another valid instrument.

RESPONSE 1:

Dear reviewer, thank you for this important comment. We see your point that the conclusion does not fully reflect the methodology. We revised the conclusions as suggested.

See page 11, line 358

In conclusion, the French version of WORQ, WORQ-BRIEF, and the clinical subscores are reliable and well targeted measures of work-related functioning in MSK. WORQ can be safely used in clinical practice to reliably measure change and comprehensively identify problems in work-related functioning. The clinical subscores may help to guide clinical decision making and intervention planning in VR or occupational health.

Comment 2:

In addition, this validation study will also benefit from assessing the agreement of WORQ and WORQ-BRIEF with another valid instrument.

RESPONSE 2:

We agree. WORQ-French  and WORQ-French Brief were compared to other valid instruments in the cross-cultural adaptation of WORQ to WORQ-French.

REF: Finger, M.; Roten-Wicki, V.; Leger, B.; Escorpizo, R., Cross-cultural adaptation of the Work Rehabilitation Questionnaire (WORQ) to French: a valid and reliable instrument to assess work functioning. J Occup Rehabil 2019 29(2):350-360. doi: 10.1007/s10926-018-9795-5.   

To provide the additional information on the external validity of WORQ-French we entered one sentence in the introduction.

Pg. 2, line 80

WORQ showed good construct validity with General Functioning (Pearson correlation: 0.66), General Health (Pearson correlation: 0.48) and HADS, (Pearson correlation: 0.56).

Reviewer 2 Report

Dear Authors,

Your study on the scale reliability and validity of WORQ, WORQ-BRIEF in French version is very interesting and good paper. The paper is generally well structured and I only have some minor comments that authors should address before publication in the Journal.

              Authors

              Line 9 - Correspondence: This word is written twice

             Abstract

              Line 11 - essential process: I suggest to write key process

               Line 12 – targeted by: I suggest to write considered within

             Introduction

            Line 30 -  Work participation:  is an major indicator with respect to which ones? I suggest              to explain better and to indicate  the reference.

             Line 32 -  psychological well-being :  I suggest to write  social and psychological benefits

enhances psychological well-being and self-worth, self-worth: I suggest to add personal identity, quality of life

Line 34 -efficiency : of what?  I suggest to write  high work performance

Line 36 - disability pension payment: I suggest to write on long-term disability allowances

Line 37 - societal cost: I suggest to write labour cost

Line 38 - social security authorities: what do you mean? Maybe national systems of social security

Line 40 - exiting the workforce early:  I suggest to write taking early retirement

Line 41- vocational/occupational rehabilitation: I suggest to write or vocational or occupational. For my opinion it is correct vocational rehabilitation

Line 42 - assessment of the workers' health problems and available resources is a critical first step to evaluate: For my opinion this sentence is too long. I suggest to write: assessment of the workers' health problems and available resources. It is a critical first step to evaluate

Line 43 - appropriate intervention: I suggest to write adequate intervention

Line 44- good practice guidelines and policy recommendations: Which? I suggest to indicate the reference

Line 48- in a cost-effective manner: I suggest to explain this sentence

Line 61- functioning: I suggest to write work-related functioning

Line 89 - the scale validity: I suggest to add  reliability

Materials and Methods

Line 99 -  how many participants participated? you wrote it only in the results (line 185) but I suggest to add also in this paragraph

Line 106 - convenience sampling: what do you mean? I suggest to explain better

Line 108- irrespective of their health condition: Why? In lines 99-100 you wrote “The participants with Musculoskeletal  injuries undergoing in-patient  rehabilitation were recruited” I suggest to explain better

Line 109-between 18 and 65 years old: one question “ To be eligible,  did the participants have to be employees at the moment of  vocational rehabilitation intervention?” If yes, I suggest to indicate it

Line 114- social system: what do you mean? Welfare system ?

Lines 148-149 -For the WORQ, no person was 148 entered into the calibration sample with more than one time point (T0-T4). In which? WORQ full version? WORQ –BRIEF? Both? I suggest to explain better

Results

Line193 – Table 2: In this table there is written 221 participants but in line 185 there is written  “Two hundred and one patient participated in the study” I suggest to review these data

                    Table 2 “ primary school” there is written 19% while in line 187 there is written 24% - for “college or university” there is written 21% while in line 187 there is written 24%. I suggest to review these data

                  Table 2 the sum of characteristics “injury score” (minor, moderate and severe) is 219 not 221

                    Table 2  the sum of characteristics “job severity” (light physical work, moderate, severe) is 231 not 221

Table 2: I suggest to write a note in which you explain that each characteristics is divided in cluster (e.i. first cluster is family status (single, married, separated) and the sum have to result 221.

Lines 186-190 -Fifty two percent……, 24% primary school, 24%..... I suggest to write the percentages in the same way or in letters or in numbers . For my opinion I suggest to write the percentages in numbers.

Line 192 - 26 from T1, 49 from T2, 46 from T3 and 50 from T4 : I suggest to add datasets for each point e.g. 26 datasets from T1

Discussion

Line 247 –DIF: I suggest to write  Differential Item Function (DIF)

Line 265 -MSK: I suggest to write the correct name and then acronym MSK because in text MSK is not explained

Line 283 - real change: what do you mean? I suggest to explain better

Line 329: work performance: I suggest to think if it is better to write work ability

Line 326: public health in the context of prevention: I suggest to write or public health or in the context of prevention – prevention activities? Health prevention?

Line 327 sustainable employment: what do you mean? I suggest to explain better

Line 329  premature drop out: I suggest to write early exit from labour market

Line 336: premature drop out from work: I suggest to write early exit from labour market

Conclusions

Line 338-339: robust measures: what do you mean? I suggest to explain better

Line 340: problems: which?

Line 340: intervention planning: which?

Line 341: Transformation tables: which? Tables 2, 3 and 4?

Line 343: may be more practical: what do you mean? I suggest to explain better

Line 345: distinct interest: I suggest to write with a specific focus

Line 346: prevention: I suggest to write prevention activities

Reference

I suggest to write Available online when you write link of website  i.e. Waddell, G.; Burton, K. Is Work good for your Health and Well-Being? Available online: www.tsoshop.co.uk (accessed  12.09).

I suggest to write the doi in each reference

Line 359 12.09 in this date It is missing the year

Line 361 It is missing the accessed to the link

Line 409 (accessed 03.19) in this date It is missing the year

Line 454 It is missing the accessed to the link

Author Response

Dear reviewer,

Thank you for reviewing our manuscript. Your comments helped to improve the paper.

Reviewer #2:

Dear Authors,

Your study on the scale reliability and validity of WORQ, WORQ-BRIEF in French version is very interesting and good paper. The paper is generally well structured and I only have some minor comments that authors should address before publication in the Journal.

Comment 3:

Line 9 - Correspondence: This word is written twice

RESPONSE 3:

Thank you for this remark, we deleted one “correspondence”

See page 1, line 9

Comment 4:    Abstract

Line 11 - essential process: I suggest to write key process

Line 12 – targeted by: I suggest to write considered within

RESPONSE 4:

As suggested we changed “essential process” to “key process” and “targeted by to “considered within”

See page 1, line 11 and line

Comment 5:      Introduction

Line 30 -  Work participation:  is an major indicator with respect to which ones? I suggest  to explain better and to indicate  the reference.

RESPONSE 5:

Thank you for this comment. We see the vagueness in this statement. To clarify our intention we rephrased the sentence.

See page 1, line 30

Work participation is considered as a major indicator of social participation for persons in working age [1-2].

As recommended, we added two References:

1.            Kunze, L.; Suppa, N. Bowling Alone or Bowling at All? The Effect of Unemployment on Social Participation EconPapers [Online], 2014. https://EconPapers.repec.org/RePEc:zbw:rwirep:510.

2.            Brand, J. E.; Burgard, S. A., Job Displacement and Social Participation over the Lifecourse: Findings for a Cohort of Joiners. Soc Forces 2008, 87 (1), 211-242.

 Comment 6:

Line 32 -  psychological well-being :  I suggest to write  social and psychological benefits enhances psychological well-being and self-worth, self-worth: I suggest to add personal identity, quality of life

RESPONSE 6:

As suggested we added “personal identity” and “quality of life” to the paragraph. We also added one Ref

See page 5, line 3

In general, and particularly for those with disabilities, work is associated with economic self-sufficiency, enhances psychological well-being and self-worth, personal identity and quality of life [3]. Work participation also assures social integration.

3.            Merchant, J. A.; Kelly, K. M.; Burmeister, L. F.; Lozier, M. J.; Amendola, A.; Lind, D. P.; KcKeen, A.; Slater, T.; Hall, J. L.; Rohlman, D. S.; Buikema, B. S., Employment status matters: a statewide survey of quality-of-life, prevention behaviors, and absenteeism and presenteeism. Journal of occupational and environmental medicine 2014, 56 (7), 686-98.

 Comment 7:

Line 34 -efficiency : of what?  I suggest to write  high work performance

Line 36 - disability pension payment: I suggest to write on long-term disability allowances

Line 37 - societal cost: I suggest to write labour cost

Line 38 - social security authorities: what do you mean? Maybe national systems of social security

Line 40 - exiting the workforce early:  I suggest to write taking early retirement

Line 41- vocational/occupational rehabilitation: I suggest to write or vocational or occupational. For my opinion it is correct vocational rehabilitation

Line 42 - assessment of the workers' health problems and available resources is a critical first step to evaluate: For my opinion this sentence is too long. I suggest to write: assessment of the workers' health problems and available resources. It is a critical first step to evaluate

Line 43 - appropriate intervention: I suggest to write adequate intervention

RESPONSE 7:

We read the comments of the reviewer and agree that changes in our wording would improve the readability of the text. We changed the formulations as suggested or refined the current wording.

See page 1, line  

Line 35 – “efficiency”  was changed to “ high work performance

Line 38 – “disability pension payment” was changed to “long-term disability allowances

Line 39 – “societal cost” was refined to “social security costs”

Line 40 – “social security authorities” was refined to “governmental social security systems

Line 42 – “exiting the workforce early” was changed to “early retirement”

Line 43 - vocational/occupational rehabilitation” as suggested, we deleted “occupational”.

 Comment 8:

Line 42 - assessment of the workers' health problems and available resources is a critical first step to evaluate: For my opinion this sentence is too long. I suggest to write: assessment of the workers' health problems and available resources. It is a critical first step to evaluate

Line 43 – “appropriate intervention” was changed to “adequate intervention”

RESPONSE 8:

Thank you for this suggestion. We have decided to revise the two long sentences in question as a whole

See page 1, line 43-48

A careful assessment of the workers' health problems and available resources is critical. in disability management, of which vocational rehabilitation is often a key part.  Research confirms a favourable return-to-work outcome by actively involving the patient in the process, e.g. evaluating his or her expectations, formulating goals and selecting interventions.

 Comment 9:

Line 44- good practice guidelines and policy recommendations: Which? I suggest to indicate the reference

RESPONSE 9:

Thank you for this comment. We already provided the following references in the manuscript:

13.       OECD, H. M. The next generation of health reforms: ministerial statement.; Paris, 2017. http://www.oecd.org/health/ministerial/ministerial-statement-2017.pdf  (accessed 2019.05.20).

14.       Multidisciplinary Association for Spinal Cord Injury Professionals (MASCIP). MASCIP - Vocational Rehabilitation Guidelines 2017. https://www.mascip.co.uk/wp-content/uploads/2017/11/Mascip-vocational-rehab-guidelines-Sept-2017.pdf (accessed 2019/05/21).

We added a new references to support the statement:

13.       Competitive Integrated Employment Services (CIES), Vocational Rehabilitation Supportive Independent Living Services (VR/IL). Vocational Services Utilization Guide. Massachusetts Rehabilitation Commission, Ed. Commonwealth of Massachusetts: 2017. https://www.mass.gov/service-details/mrc-vocational-rehabilitation-guidelines    (accessed 2019/07/15)

 Comment 10:

Line 48- in a cost-effective manner: I suggest to explain this sentence

RESPONSE 10:

Thank you for this comment. We revised the sentence and added the requested information.

See page 2, line 54

…  - in a cost-effective manner – compared to clinical testing of work functioning [16-19].

19.       Kuijer, P. P.; Gouttebarge, V.; Wind, H.; van Duivenbooden, C.; Sluiter, J. K.; Frings-Dresen, M. H., Prognostic value of self-reported work ability and performance-based lifting tests for sustainable return to work among construction workers. Scand J Work Environ Health 2012, 38 (6), 600-3.

Comment 11:

Line 61- functioning: I suggest to write work-related functioning

Line 89 - the scale validity: I suggest to add  reliability

RESPONSE 11:

We changed the wording as suggested by the reviewer.

See page 2

Line 67 – “functioning”  was changed to “ work-related functioning

Line 94 – “reliability” was added to the sentence

Materials and Methods

Comment 12:

Line 99 -  how many participants participated? you wrote it only in the results (line 185) but I suggest to add also in this paragraph   (221)

          RESPONSE 12:

As suggested, we added the number of participants into the first sentence of the of the methods section.

          See page 3, line 109

We collected the data from two hundred and twenty-one participants in a longitudinal observational study.

 Comment 13:

Line 106 - convenience sampling: what do you mean? I suggest to explain better

          RESPONSE 13:

Thank you for this comment. We added some information to the text to better characterize the sample.

See page 3, line 114

Participants were recruited consecutively using convenience sampling, i.e. all patients were checked for their inclusion criteria when they entered the Department of Vocational Rehabilitation.

 Comment 14:

Line 108- irrespective of their health condition: Why? In lines 99-100 you wrote “The participants with Musculoskeletal  injuries undergoing in-patient  rehabilitation were recruited” I suggest to explain better

          RESPONSE 14:

Thank you for this comment. The reviewer is correct, all participants were referred to the VR department because of a MSK diagnosis. “irrespective of their health condition” means, that we included participants irrespective of the location of the injury. We clarified this issue in the manuscript.

See page 3, line 117

          … irrespective of their health type of MSK condition.

Comment 15:

Line 109-between 18 and 65 years old: one question “ To be eligible,  did the participants have to be employees at the moment of  vocational rehabilitation intervention?” If yes, I suggest to indicate it.

          RESPONSE 15:

Thank you for this question. No, the participants in this study didn’t have to be employed. They had to be employed or looking for a new job.  Persons with a disability pension were not eligible to the department of VR and therefore excluded from our study sample. We added this information to the inclusion criteria.

See page 3, line 117

(3) were between 18 and 65 years old, (4) employed or looking for a new job

 Comment 16:

Line 114- social system: what do you mean? Welfare system ?

RESPONSE 16:

Thank you for this remark. We realized that the different organization and naming of the national social security or health care systems may be confusing to the reader. In Switzerland, the national disability found (IV) is paying for VR interventions and in this time for weekly allowances. IV is also paying a lifelong disability rent for persons with a permanent disability. Welfare benefits are payed for non-disabled persons in need.

Therefore we decided to change the term “social system” to “labour and employment services” 

See page 3, line 124

 Comment 17:

Lines 148-149 -For the WORQ, no person was 148 entered into the calibration sample with more than one time point (T0-T4). In which? WORQ full version? WORQ –BRIEF? Both? I suggest to explain better

RESPONSE 17:

The calibration sample was constructed based on the dataset of the full version of WORQ. As WORQ-BRIEF and the clinical sets are subsets of WORQ, all analysis were based on the calibration sample.

See page 5, line 148

Initially, we constructed a dataset called “calibration sample” that contained the 40 questions from part 2 of the full version of WORQ.

line 148

The calibration sample was used to analyse WORQ, WORQ-BRIEF and the clinical subscores.

Results

Comment 18: Table 2

a) Line193 – Table 2: In this table there is written 221 participants but in line 185 there is written  “Two hundred and one patient participated in the study” I suggest to review these data.                   

b) Table 2 “ primary school” there is written 19% while in line 187 there is written 24% - for “college or university” there is written 21% while in line 187 there is written 24%. I suggest to review these data.

c) Table 2 the sum of characteristics “injury score” (minor, moderate and severe) is 219 not 221.

d) Table 2  the sum of characteristics “job severity” (light physical work, moderate, severe) is 231 not 221.

e) Table 2: I suggest to write a note in which you explain that each characteristics is divided in cluster (e.i. first cluster is family status (single, married, separated) and the sum have to result 221.

 RESPONSE 18:

a) Thank you for this comment. There was a mistake in the text. We revised the text to “Two hundred and twenty one participant. The “twenty” was missing.

b) The 24% in the text are referring to “primary school or less” therefore we counted  “less than primary school” 5% and “primary school” 19% together.
The 24% in the text referred to college or university education (9%) and post graduate education (3%). We see that this is not clear in the text, therefore we reworded the sentence.

See page 5, line 200

…24% reported to have a college or university degree including post graduate education.

c) We see the point. We didn’t include the 3 missing values in the table. We revised table 2 and added the missing values.

d) Thank you for this remark. We realized that we reported the wrong data under “Job severity”. We corrected the table.

e) Thank you for this suggestion. We realized that due to the formatting of the table it was difficult to see the cluster. We formatted the table now according to the cluster to clarify the data structure.

Characteristics

n=221 (%)

Age,   mean, years

43.47 (10.9 SD)

Sex,   male

196   (89.0%)

Family status

single

married   / cohabitant

separated / divorced

  48 (21.7%)

125   (56.5%)

  48 (21.7%)

Injury location

upper   extremity

lower   extremity

trunk/back

poly-trauma

85   (38.6%)

82   (37.1%)

25   (11.4%)

29   (13.2%)

Injury score

minor

moderate

severe

missing

  58 (26.2%)
  120 (54.3%)

  41 (18.6%)

  3 ( 1.4%)

Job severity

light   physical work

moderate   physical work

severe physical work

 46 (20.8%)

103   (46.6%)

 72 (32.9%)

Education

less   than primary school

primary   school

secondary   school

college   or university

post graduate education

 11 ( 5%)

 41 (19%)

115   (52%)

 47 (21%)

  7 ( 3%)

 Comment 19:

Lines 186-190 -Fifty two percent……, 24% primary school, 24%..... I suggest to write the percentages in the same way or in letters or in numbers . For my opinion I suggest to write the percentages in numbers.

 RESPONSE 19:

Thank you for this comment. We agree that writing the percentages in numbers is better to understand. We revised the sentence accordingly.

See page 5, line 189

52% of the participants had finished high school ….

 Comment 20:

Line 192 - 26 from T1, 49 from T2, 46 from T3 and 50 from T4 : I suggest to add datasets for each point e.g. 26 datasets from T1

RESPONSE 20:

Thank you for this comment. The abbreviation (MSK) was introduced in the methods section page 3, line 107. But we realized that the abbreviation (MSK) is not familiar therefore we

Discussion

Comment 21:

Line 247 –DIF: I suggest to write  Differential Item Function (DIF)

RESPONSE 21:

As suggested by the reviewer, we wrote the full name “Differential Item Function” instead of the abbreviation (DIF).

See page 9, line 261

 Comment 22:

Line 265 -MSK: I suggest to write the correct name and then acronym MSK because in text MSK is not explained

RESPONSE 22:

Thank you for this comment. The abbreviation (MSK) was introduced in the methods section page 3, line 107. But because we realized that the abbreviation (MSK) is not familiar we added the full name as suggested.

See page 9, line 279

 Comment 23:

Line 283 - real change: what do you mean? I suggest to explain better

RESPONSE 23:

Thank you for this comment. We deleted “real” and explained the characteristics of an interval scale in an additional sentence.

See page 9, line 297

This finding indicates a much smaller real change on the interval scale compared to the ordinal raw score. Interval scales are characterized by measurement units that maintain the same size across the entire scale, resulting in a more accurate measurement of change than an ordinal scale.

 Comment 24:

Line 329: work performance: I suggest to think if it is better to write work ability

RESPONSE 24:

          We agree and changed the term “work performance” to “work ability”

See page 11, line 349

 Comment 25:

Line 326: public health in the context of prevention: I suggest to write or public health or in the context of prevention – prevention activities? Health prevention? Line 327 sustainable employment: what do you mean? I suggest to explain better
Line 329  premature drop out: I suggest to write early exit from labour market
Line 336: premature drop out from work: I suggest to write early exit from labour market

RESPONSE 25:

Thank you for this comment. We decided to revised the whole paragraph to clarify the ambiguities. We deleted the two last sentences as they were redundant.

See page 10, line 341

Based on this finding, it should be further evaluated whether WORQ could also be used in public health in the context of health prevention and to support sustainable employment. Sustainable employment characterizes a person – job – workplace match that enables a person to stay healthy and satisfied at work over time, with a work performance that meets the expectations of the person and the employer [62]. A focus on prevention of work related disability is becoming increasingly important, as an increasing number of workers experience a decline in work-related functioning while aging. An instrument that detects change in functioning early, may help to identify those who need support and to prevent reduced work ability or early exit from labour market.

Conclusions

Comment 26:

Line 338-339: robust measures: what do you mean? I suggest to explain better

RESPONSE 26:

Thank you for this comment. We changed the term “robust” to “reliable and well targeted

See page 11, line 359

Comment 27:

Line 340: problems: which?

Line 340: intervention planning: which?

RESPONSE 27:

          We revised the sentence to clarify the information.

See page 11, line 359

WORQ can be safely used in clinical practice to comprehensively identify problems in work-related functioning, guide clinical decision making and intervention planning in VR or occupational health.

Comment 28:

Line 341: Transformation tables: which? Tables 2, 3 and 4?

 RESPONSE 28:

Thank you for this comment. The transformation tables for WORQ, WORQ-BRIEF and the clinical subscores are reported in the supplementary material. (see Table S3). This information can be found on page 9, line 248.

To clarify this information, we added (see Table S3) to the sentence in the conclusion section.

See page 11, line 361

Transformation tables (Table S3) enable the calculation of change scores to document reliable change in functioning,..

 Comment 29:

Line 343: may be more practical: what do you mean? I suggest to explain better

 RESPONSE 29:

Thank you for this comment. We followed the suggestion and added some information to clarify the meaning of the sentence.

See page 11, line 361

With only 13 items, WORQ-BRIEF may be more practical to be used in research or to compare populations across health conditions compared to WORQ.

 Comment 30:

Line 345: distinct interest: I suggest to write with a specific focus

Line 346: prevention: I suggest to write prevention activities

RESPONSE 30:

We agree and changed the wording of the sentence. We changed “distinct interest” to “specific focus” and we added “activities”

See page 11, line 366

with a distinct interest specific focus in populations with mental problems or multiple comorbidities and in prevention activities.

Reference

Comment 31:

I suggest to write Available online when you write link of website  i.e. Waddell, G.; Burton, K. Is Work good for your Health and Well-Being? Available online: www.tsoshop.co.uk (accessed  12.09).

I suggest to write the doi in each reference

RESPONSE 31:

Thank you for the comments. We added the Available online: to all online documents. And we added the doi to each journal reference.

 Comment 32:

Line 359 12.09 in this date It is missing the year

Line 361 It is missing the accessed to the link

Line 409 (accessed 03.19) in this date It is missing the year

Line 454 It is missing the accessed to the link

RESPONSE 31:

We checked the dates and added the missing access to the link for the references

See page 12, line 391 and 393

See page 12, line 421

See page 13, line 468
